# ZERO-TO-INTERACTION: GENERATING DYNAMIC VIDEOS FROM SYNTHETIC STATE TRANSITIONS

## ABSTRACT

While recent video generative models can synthesize high-fidelity videos, they struggle to portray plausible physical interactions and the resulting state transitions, a critical bottleneck for applications in robotics and VR/AR. To address this, we introduce a framework to generate a scalable synthetic dataset of controllable interactions. Our pipeline leverages a structured taxonomy and state-of-the-art image editing models to create explicit 'start' and 'end' state images, which serve as visual anchors for the interaction. To generate a seamless video utilizing these anchors, we propose State-Guided Sampling (SGS), a novel sampling technique that mitigates artifacts common in naive conditional generation. Furthermore, we develop and validate a new automated evaluation system that aligns with human judgments to ensure data quality. Experiments show that fine-tuning a base model on our dataset significantly enhances its ability to generate plausible interactions. The dataset, code, and evaluation tools will be released. Project page is available at `https://zero2interaction.github.io/`.

## 1 INTRODUCTION

Recent advances in video generative models (Ho et al., 2022; Kong et al., 2024; Wan et al., 2025) have enabled the synthesis of high-fidelity videos. This capability has led to growing exploration of their use in applications, e.g., world models for robotics (Agarwal et al., 2025) and developing immersive content for virtual and augmented reality (VR/AR) such as 4D content generation (Wu et al., 2025; Liu et al., 2025b). Despite their ability to produce visually realistic samples, these models struggle to accurately capture plausible physical interactions and the resulting state transitions among objects. For instance, without a reliable model of physical interaction and the state dynamics, a world model cannot effectively guide a robot in learning object manipulation, nor can it support the creation of dynamic scenarios essential for user engagement in VR/AR environments.

To address this issue, some studies have incorporated auxiliary conditions, such as segmentation maps (Akkerman et al., 2025) or used large language models (LLMs) to improve physical fidelity (Xue et al., 2025; Zhang et al., 2025a). Others have focused on fine-tuning models on newly collected, large-scale human-object interaction datasets (Liu et al., 2025a).

Nevertheless, methods using auxiliary conditions lack generalizability, while data-driven approaches are constrained by the limited scope of existing datasets. Manually curating large-scale datasets is prohibitively costly, making synthetic data a viable alternative. However, generating high-quality, diverse, and physically plausible synthetic interactions remains an open challenge.

In this work, inspired by the recent success of training LLMs with synthetic data (Li et al., 2024a; Zhao et al., 2025), we propose a framework for constructing a dataset to improve the capacity of video generative models in synthesizing physically plausible interactions and subsequent object state transitions as shown in Figure 1. Specifically, we first define a taxonomy for generating prompts that describe plausible interactions, comprising attributes such as the interactable object and the type of state transition-oriented interaction. However, we identified a key challenge in directly applying these prompts to video generation models, as the resulting outputs often fail to accurately depict the intended interactions.

Therefore, we incorporate an intermediate image generation step to ensure fidelity. Instead of direct text-to-video synthesis, we first generate an initial image from the prompt. Following that, a state-of-

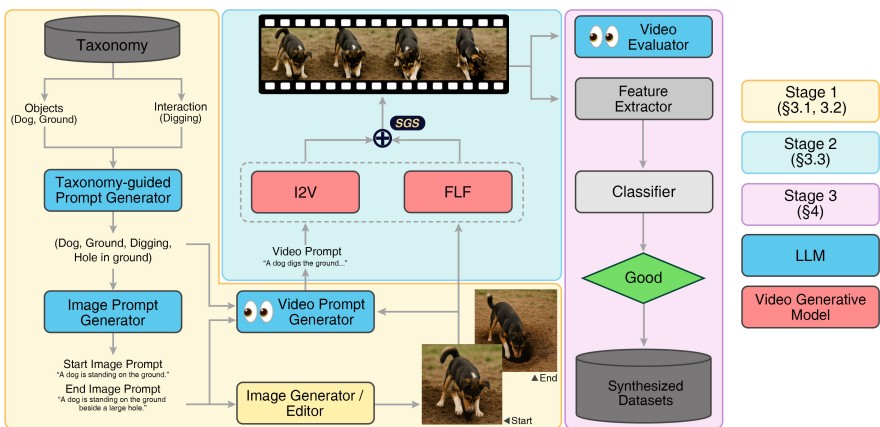

Figure 1: **Overview of Interaction-Centric Video Dataset Generation Pipeline.**

the-art image editing model alters this image to reflect the object's subsequent state change according to the given interaction. The video generation is then conditioned on this pair of images, which clearly defines the start and end states of the interaction.

However, we observe that relying solely on the pair for the first-to-last frame generation often leads to undesirable artifacts, including abrupt scene changes or unnatural transformations that compromise the video's temporal consistency. To address this challenge, we propose State-Guided Sampling (SGS), a novel sampling method designed to guide the model toward a smooth and plausible state transition.

To validate our approach, we develop an automated system to evaluate interaction quality, confirming its reliability against human judgments. Our experiments show that a model fine-tuned on our curated dataset significantly enhances its capability to generate complex interactions. Our main contributions are as follows:

- We introduce a novel pipeline to construct a synthetic dataset for diverse object interactions, based on a structured taxonomy and state-of-the-art image editing models that create explicitly 'start' and 'end' state images.

- We propose State-Guided Sampling (SGS), a novel sampling strategy that mitigates visual artifacts and guides video models to generate seamless state transitions.

- We develop a model-based evaluation system to assess interaction quality in generated videos, and validate its alignment with human judgments to ensure data quality and facilitate scalable dataset curation.

- We release the full dataset, data generation pipeline, and evaluation tools to the public to facilitate future research.

## 2 RELATED WORKS

### 2.1 VIDEO GENERATIVE MODEL

Early diffusion-based video generative models evolved from U-Net-based image architectures by incorporating separate temporal modules (Ho et al., 2022; Blattmann et al., 2023; Guo et al., 2023; Xing et al., 2023; Hong et al., 2022). The subsequent emergence of the Diffusion Transformer (DiT) architecture (Peebles & Xie, 2023) unified spatial and temporal modeling into a single self-attention backbone (Ma et al., 2024). This approach demonstrated superior scalability in both model parameters and training data, enabling the modeling of more complex temporal dynamics. Building on the DiT architecture, a new wave of models capable of generating high-quality video has been introduced, including prominent examples like SoRA, Hunyuan, and Wan 2.1 (Liu et al., 2024b; Zheng et al., 2024; Yang et al., 2024; Kong et al., 2024; Wan et al., 2025). While these models excel at generating high-fidelity videos from general text conditions, they often exhibit limitations

in accurately modeling out-of-distribution prompts (Xue et al., 2025) such as depicting physical laws and interactions.

## 2.2 GENERATING DYNAMIC INTERACTIONS IN VIDEOS

Prior works on interaction generation face trade-offs between controllability and generality. For example, InterDyn (Akkerman et al., 2025) uses segmentation maps for fine-grained control, which limits its applicability. Similarly, HOIGen (Liu et al., 2025a) is constrained to the human-object interaction domain. Another line of research focuses on injecting knowledge into the models. Approaches like PhyT2V (Xue et al., 2025) utilize MLLMs to iteratively refine prompts, but this does not enhance the internal capabilities of the generative model itself. DiffPhy (Zhang et al., 2025a) enhances training prompts with physically-grounded descriptions from an LLM, while VideoREPA (Zhang et al., 2025b) injects physical knowledge via representation matching from a video encoder. However, these methods are fundamentally oriented towards acquiring knowledge from real-world data, thus limiting their capacity to generate novel and creative interaction scenarios.

In contrast, our work decouples the generation process from the dependency on existing video datasets. We achieve this by introducing a new form of general-purpose controllability, which utilizes 'start' and 'end' images as result-driven visual anchors to precisely define an interaction's outcome. This novel control mechanism is the key that enables our scalable, synthetic dataset generation framework, capable of generating high-quality, novel, and creative interactions.

## 2.3 SYNTHETIC DATASET GENERATION

While the performance of large-scale generative models relies heavily on the amount of high-quality data, the growing challenges of collecting and curating real-world datasets have made the use of synthetic data an increasingly essential research direction. Early works leveraged synthetic data to induce novel capabilities; LLaVa (Liu et al., 2023) used image metadata to create visual instruction-tuning data, and InstructPix2Pix (Brooks et al., 2023) built datasets by refining ambiguous prompts. The approach employing LLM has been further advanced through self-improvement methods that enhance performance via self-correct (Liu et al., 2024a; Welleck et al., 2022) and filtering (Li et al., 2024b). Building on these ideas, subsequent works like GLAN (Li et al., 2024a) and Absolute-Zero (Zhao et al., 2025) have even eliminated the need for initial seed data. Inspired by this line of research, we propose a framework that generates and filters a synthetic dataset specialized for interaction and state change. Our framework uniquely leverages image generation and editing models, in conjunction with our State-Guided Sampling method, to create high-quality, targeted training data.

## 3 SYNTHETIC INTERACTION VIDEO GENERATION

To generate high-fidelity videos of complex interactions, we propose a multi-stage data generation pipeline. Our approach is designed to overcome the limitations of direct text-to-video synthesis by ensuring both semantic accuracy and temporal consistency. The pipeline consists of three main stages: (1) structured prompt generation based on a custom taxonomy focusing on state transitions, (2) synthesis of 'start' and 'end' state image pairs using state-of-the-art image editing, and (3) temporally coherent video generation guided by the proposed State-Guided Sampling (SGS) technique. Figure 1 overviews our proposed dataset generation pipeline, where each stage is designed to address key challenges in generating plausible object interactions, as detailed in the following subsections.

### 3.1 TAXONOMY-GUIDED PROMPT GENERATION

To generate a diverse yet plausible set of interaction scenarios, we first address the limitations of a naive approach of directly querying a Large Language Model (LLM). This method often results in a strong bias towards common, high-frequency interactions (e.g., *a person holding a cup*) and fails to cover creative or rare cases.

To overcome this limitation, we introduce a structured prompt generation process founded on a purpose-built taxonomy inspired by ImageNet (Deng et al., 2009). Our taxonomy is designed for comprehensive coverage, encompassing approximately 1,300 objects and 500 interaction types ranging from mundane to imaginative concepts. Notably, the interactions are semantically organized by

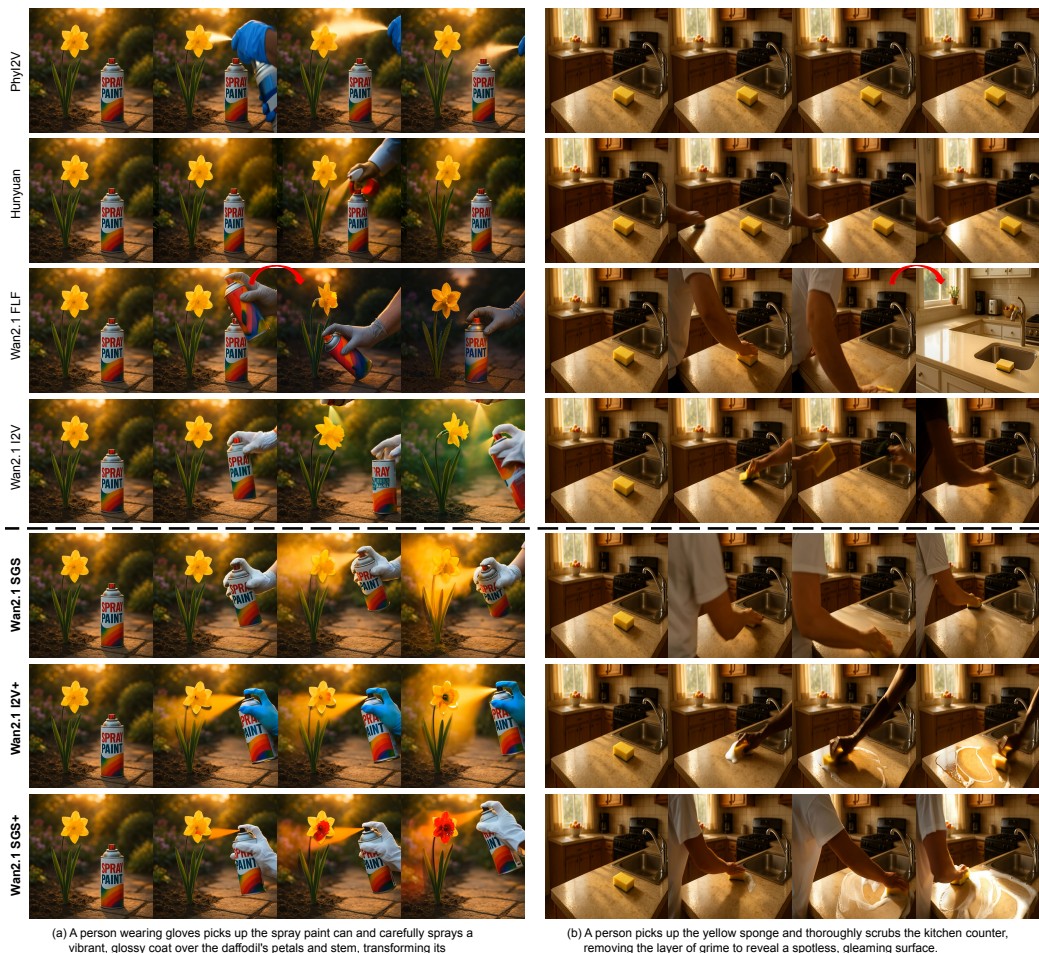

(a) A person wearing gloves picks up the spray paint can and carefully sprays a vibrant, glossy coat over the daffodil's petals and stem, transforming its appearance in the evening garden.

(b) A person picks up the yellow sponge and thoroughly scrubs the kitchen counter, removing the layer of grime to reveal a spotless, gleaming surface.

Figure 2: **Qualitative Evaluation.** The red arrow indicates abrupt frame changes, while the plus (+) denotes our fine-tuned model.

their physical outcomes (e.g., *Deformation, Separation/Fracture*), providing a systematic basis for generating meaningful state transitions.

Our process employs two complementary sampling strategies, object-centric and interaction-centric. For object-centric sampling, we randomly select a pair of objects from our taxonomy and use an LLM as a plausibility filter to determine if a feasible interaction can be conceived between them. If validated, the LLM generates the corresponding interaction and resulting state change. This allows for the discovery of novel and unexpected yet plausible scenarios. Conversely, interaction-centric sampling focuses on generating diverse object combinations for a single type of interaction. To maintain scalability, we manage a localized context for each interaction type, enabling the LLM to efficiently explore varied object pairings without repetition. This dual-strategy process yields a structured tuple of the form (`object1`, `object2`, `interaction`, `state change`), which serves as the semantic foundation for the visual synthesis stage. Further details on the construction and scope of our taxonomy are provided in Appendix B.

## 3.2 INTERACTION-STATE IMAGE PAIR SYNTHESIS

While the structured tuple provides a clear semantic description, direct text-to-video synthesis often fails to faithfully render the specified interaction and its precise state transition. To address this, we adopt an image-to-video framework that relies on explicit visual conditions. The generated tuple in Section 3.1 is used to synthesize a pair of images: an initial 'start' image depicting the scene prior to the interaction, and a corresponding 'end' image reflecting the object's state change.

To synthesize a sample, we first use the tuple to compose a detailed prompt for a text-to-image model to generate the initial frame. Subsequently, the `state change` component of the tuple guides a state-of-the-art image editing model to modify the initial image into the final frame. This resulting image pair provides strong visual anchors that explicitly define the start and end points of the interaction for the video generation model. We use GPT-4o (Hurst et al., 2024) for prompt generation, image creation, and editing.

## 3.3 STATE-GUIDED SAMPLING

Given the 'start' (first-frame) and 'end' (last-frame) images, the next stage is to synthesize a video that plausibly and seamlessly connects these two states while representing the prompt. The primary challenge is balancing global guidance toward the end state with local, frame-to-frame temporal coherence. An I2V model (start-frame conditioned) provides local coherence but lacks global direction, whereas an FLF model (start-and-end-frame conditioned) has strong global guidance but can produce artifacts. To resolve this trade-off, we introduce State-Guided Sampling (SGS), a novel sampling technique that dynamically combines the velocity fields of an I2V model ($v_I$) and an FLF model ($v_F$) within a flow-matching framework (Lipman et al., 2022). Our final velocity field, $v_{sgs}$, is defined as a dynamic, frame-wise weighted sum of the two as follows:

$$v_{\text{sgs}}(z_t, t, c') = (\mathbf{1} - W) \odot v_F(z_t, t, c') + W \odot v_I(z_t, t, c), \tag{1}$$

where $\odot$ denotes the frame-wise product, $W$ is the frame-wise weight. $c$ represents the condition for the prompt and start image, and $c'$ further includes the last image. We find that a naive approach of linearly interpolating the velocity fields creates a 'ghosting effect' as shown in Figure 9, which is a translucent overlay of the start and end states where the FLF model's rigid guidance conflicts with local temporal consistency. To alleviate this effect, we design a dynamic frame-wise weighting scheme where the weight $W_f$ for each frame index $f$ to smoothly transit the model's reliance from global guidance to local coherence, calculated using a normalized exponential curve:

$$W_f = \alpha + (\beta - \alpha) \cdot \frac{e^{k \cdot \frac{f}{F-1}} - 1}{e^k - 1} \tag{2}$$

where $F$ is the total number of frames. The interpolation begins with a starting weight $\alpha$ to ensure an initial blend of both models and concludes with $\beta = 1.0$, allowing the I2V model to dominate the final frames for a coherent finish. The parameter $k$ controls the curve's steepness, determining how long the FLF model's stronger state guidance is maintained. We set $\alpha = 0.5$ and $k = 5.0$ in our experiments. In essence, SGS resolves the conflict between global and local objectives by initially utilizing the FLF model's trajectory guidance and then gradually shifting to the I2V model's strength in ensuring visual consistency, ultimately producing a plausible and seamless state transition.

## 4 INTERACTION QUALITY ASSESSMENT

Holistically evaluating whether a generated video contains plausible interactions and state transitions is crucial, yet a standardized metric is currently lacking. Existing metrics are often limited to specific domains and fail to capture semantic plausibility. Therefore, we identify the need for a comprehensive framework capable of assessing the diverse aspects of quality of generated dynamic events. This section introduces our proposed hybrid evaluation framework, which combines the semantic understanding of a large Vision-Language Model (VLM) with specialized, auxiliary features to ensure robust and reliable assessment.

### 4.1 CRITERIA FOR INTERACTION QUALITY

To overcome the limitations of prior metrics, we first define a framework that assesses video quality across four key criteria. The first, *Interaction Presence & Clarity*, evaluates whether the specified interaction occurs and is unambiguously depicted in the video. The second criterion, *Interaction-State Causality*, assesses if the object's state transition is a direct and causal consequence of the specified interaction. The third, *Physical Plausibility*, determines if the object's motion and the interaction's outcome adhere to physical principles. The last criterion, *Temporal Continuity*, checks for a smooth and consistent flow, free of visual artifacts such as dissolves, scene cuts, or distortions.

Table 1: **Quantitative Evaluation.** SA and PC denote Semantic Adherence and Physical Commonsense, respectively. The **bold** represents the best, and the underline does the second best.

| Model | | VLM-Assisted Score (↑) | | | | | Temporal Artifact (↓) | VideoPhy2 (↑) | |
|---|---|---|---|---|---|---|---|---|---|
| | | Clarity | Causality | Plausibility | Continuity | Average | | SA | PC |
| PhyI2V[1] | | 2.52 | 2.34 | 2.44 | 3.40 | 2.68 | 0.44 | 0.26 | 0.58 |
| HunyuanVideo | | 1.71 | 1.50 | **3.06** | **4.16** | 2.61 | 0.09 | 0.25 | **0.77** |
| Wan 2.1 | FLF | 3.10 | 3.06 | 2.87 | 3.20 | 3.06 | 0.68 | 0.31 | 0.56 |
| | I2V | 2.90 | 2.86 | 2.98 | 3.87 | 3.15 | 0.13 | 0.30 | 0.56 |
| | SGS | 3.00 | 2.94 | 2.91 | 3.96 | 3.20 | 0.12 | 0.26 | 0.53 |
| Wan 2.1 | I2V | 3.20 | 3.05 | 2.92 | 3.96 | 3.28 | **0.07** | **0.34** | 0.57 |
| (Fine-tuned) | SGS | **3.23** | **3.21** | 3.03 | 3.90 | **3.34** | 0.11 | 0.32 | 0.53 |

## 4.2 VLM-ASSISTED EVALUATION AND LIMITATIONS

We initially used a VLM (Gemini-2.5-Pro (Comanici et al., 2025)) to score videos from 1 to 5 across our four criteria, validating it against 1,146 human-annotated videos. Pearson correlation was strong for semantic criteria like *Clarity* ($\rho = 0.54$), and *Causality* ($\rho = 0.49$). but weaker for Plausibility ($\rho = 0.37$) and *Continuity* ($\rho = 0.31$). It shows that using VLM only for evaluation is not sufficient to match human judgment.

## 4.3 ENHANCED EVALUATION WITH AUXILIARY FEATURES

To enhance the overall reliability of our framework, we adopted a hybrid approach by integrating auxiliary features for compensating weaker criteria. Specifically, to augment *Physical Plausibility* and *Temporal Continuity*, we integrate two auxiliary features. These include the *Physical Commonsense* score (Bansal et al., 2025) from a specialized VLM, the *Surprising* score (Garrido et al., 2025) from a pre-trained V-JEPA 2 model. Furthermore, we incorporate predictions from our own trained Plausibility Probe (PP) and Quality Classifier (QC) as ad-

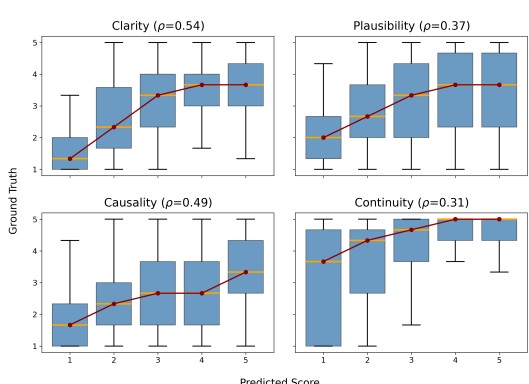

Figure 3: **VLM-Human Rating Correlation.**

Figure 4: **Ablation Study on SVM Inputs.** PP and QC denote the features from our Plausibility Probe and Quality Classifier, respectively.

| | F1 (Good) | F1 (Bad) | Macro F1 | AUC |
|---|---|---|---|---|
| VideoPhy2 | 0.00 | 0.80 | 0.40 | 0.57 |
| Base | 0.53 | 0.80 | 0.66 | 0.71 |
| (a) + Prefilter | 0.59 | 0.77 | 0.68 | 0.72 |
| (b) + PC score | 0.60 | 0.78 | 0.69 | 0.72 |
| (c) + Surprise | 0.61 | 0.79 | 0.70 | 0.73 |
| (d) + PP | 0.61 | 0.79 | 0.70 | 0.76 |
| (e) + QC | **0.63** | **0.80** | **0.71** | **0.77** |

ditional features, which are lightweight attention modules trained on V-JEPA2 features to predict the human-annotated physical plausibility scores and quality labels, respectively. Lastly, we utilize a dedicated temporal artifact detector, which is detailed in the following subsection. We experimentally verify the effectiveness of each auxiliary feature in Section 5.2.

## 4.4 TEMPORAL ARTIFACT DETECTION FOR CONTINUITY

A persistent challenge that degrades *Temporal Continuity* is the occurrence of abrupt scene transitions or dissolve artifacts in generated videos. While we initially investigated using a powerful VLM for this task, we found it was unable to reliably detect such sudden and unnatural scene changes. To this end, we employ a frozen V-JEPA 2 (Assran et al., 2025) model as a feature extractor and train a transformer-based attention classifier for a binary classification task.

To train this detector, we constructed a synthetic dataset by applying one of three distinct augmentation processes to video clips drawn from the UCF-101 dataset (Soomro et al., 2012). Each process simulates a different type of temporal artifact described as follows.

**Cross-Fade Transition** is an alpha blending transition applied between two clips, where the start time and duration of the fade are randomized.

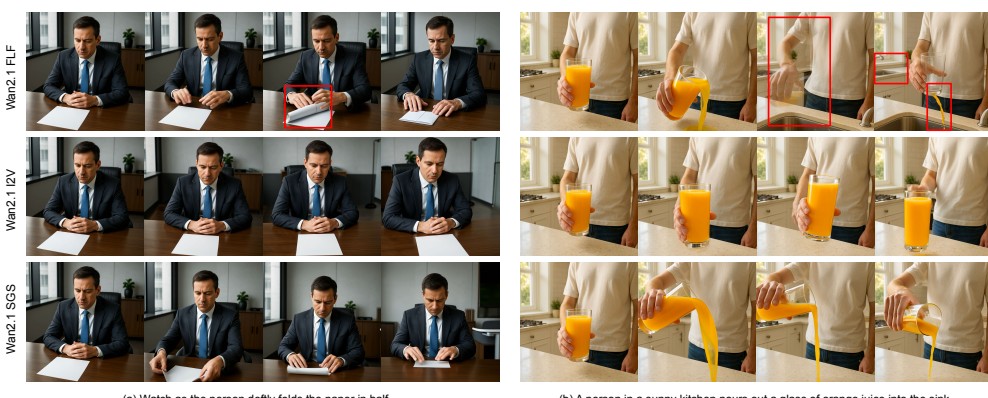

(a) Watch as the person deftly folds the paper in half.   (b) A person in a sunny kitchen pours out a glass of orange juice into the sink.

Figure 5: **Effectiveness of State-Guided Sampling.** The proposed SGS effectively resolves the unnatural scene transitions.

**Hard Cut** simulates an abrupt scene change by concatenating segments from two distinct videos at a randomized temporal midpoint without any blending.

**Intra-Scene Displacement** creates continuity errors that mimic a camera jump. A video clip is cut at a random point, the entire second segment is spatially translated, and the two segments are then rejoined with a brief cross-fade. A detailed performance evaluation is given in Appendix D.

## 5 EXPERIMENTS

To evaluate our dataset and methodology, we select the state-of-the-art open-source model, Wan 2.1 (Wan et al., 2025), as our base model for fine-tuning. Its strong performance in Image-to-Video (I2V) generation makes it a suitable foundation for validating our proposed contributions. For performance comparison, we benchmark against two strong baselines: HunyuanVideo (Kong et al., 2024) and PhyI2V[1] adapted from PhyT2V (Xue et al., 2025).

### 5.1 DATASET CONSTRUCTION DETAILS

Our synthetic dataset is built through an iterative generation and curation process. We first generate an initial pool of 1,146 videos from 191 prompts using six different sampling methods (I2V, FLF, and four SGS with varying settings of $\alpha$). This initial set undergoes detailed human annotation by the domain experts, who provide (1) a binary 'Good'/'Bad' quality label, (2) 1-5 scores for our four proposed criteria (*Clarity, Causality, Plausibility, Continuity*), and (3) a relative ranking of the generated videos per prompt. The final 'Good'/'Bad' labels are determined by majority vote, and the criterion scores are averaged. This high-quality, human-filtered data is then used to fine-tune the base model in a lightweight manner (Hu et al., 2022).

To construct the complete dataset, we employ both original and fine-tuned models to generate an additional 4,971 videos from 1,018 new prompts. This expanded set is then curated using our automated evaluation system detailed in Section 4. The final training dataset combines the initially human-filtered samples from the bootstrapping phase with the videos that successfully passed our automated filtering process. The final training set consists of 1,525 videos from 681 prompts, a volume we believe is sufficient for effective alignment (Zhou et al., 2023). Each video has 81 frames and is 5 seconds long. A held-out set of 105 prompts is used for validation.

### 5.2 RELIABILITY OF EVALUATION FRAMEWORK

To validate the reliability of the proposed evaluation framework, we use the 1,146 human-annotated videos from our bootstrapping set as the ground truth for this analysis. Our final goal for the evaluation pipeline is to train an SVM classifier (Hearst et al., 1998) on the proposed features to accurately predict the human 'Good'/'Bad' labels. We conducted an ablation study to quantify each compo-

---

[1]PhyI2V modifies PhyT2V by replacing the CogVideoX-5B with CogVideoX-5B-I2V (Yang et al., 2024).

Table 2: **VBench (Motion) and Human Evaluation.**

| | Vbench | | | Human Comparison | |
|---|---|---|---|---|---|
| | **Motion Smoothness** | **Dynamic Degree** | **Aesthetic Quality** | **Rank(↓)** | **WR** |
| PhyI2V | 0.99 | 0.57 | 0.57 | 2.90 | 8% |
| Hunyuan | 0.99 | 0.20 | 0.63 | 3.43 | 2% |
| Wan2.1 I2V | 0.98 | 0.64 | 0.63 | 1.98 | 29% |
| +Fine-tuning | 0.98 | **0.73** | 0.63 | **1.68** | **53%** |

Table 3: **Ablation Study on the Initial I2V Weight, $\alpha$, for SGS.**

| | Clarity | Causality | Plausibility | Continuity | Rank(↓) | Good | Temporal Artifact(↓) |
|---|---|---|---|---|---|---|---|
| FLF | 2.96 | 2.64 | 2.82 | 2.41 | 4.08 | 0.15 | 0.66 |
| I2V | 2.64 | 2.36 | 3.01 | 4.22 | 3.71 | 0.22 | 0.13 |
| $\alpha = 0.3$ | 3.04 | 2.72 | 3.05 | 4.21 | 3.05 | 0.30 | 0.26 |
| $\alpha = 0.4$ | 3.13 | 2.81 | **3.11** | 4.30 | **2.98** | 0.35 | 0.20 |
| $\alpha = 0.5$ | **3.14** | **2.86** | 3.07 | **4.56** | 3.05 | **0.38** | **0.11** |
| $\alpha = 0.6$ | 2.95 | 2.65 | 3.08 | 4.55 | 3.38 | 0.31 | **0.11** |

nent's contribution. While VLM scores provide a strong semantic baseline (Figure 3), they require augmentation for physical and temporal assessment. To identify the most effective feature set, we conducted a comprehensive ablation study by training an SVM classifier with cross-validation. A baseline using only a pre-existing Physical VLM Bansal et al. (2025) fails to identify any 'Good' videos, yielding an F1-score (pos) of 0.00. In contrast, using our four VLM-based scores as base features (Base) provides a much stronger starting point, achieving a Macro F1 of 0.66. To prevent temporal artifacts from confounding our analysis, the validation protocol pre-filters videos using the temporal artifact detector. As shown in Table 4(a), this step isolates the effectiveness of other features and improves the positive-class F1 score from 0.53 to 0.59.

On the cleaned dataset, we then incrementally added our auxiliary features to complement physical plausibility. The addition of the VideoPhy2 *PC* score (Table 4(b)) and the V-JEPA 2 *surprise* score (Table 4(c)) steadily increased performance. We observed further improvement by incorporating features from our Attentive Plausibility Probe (Table 4(d)), a shallow attention module added to V-JEPA 2 that is trained to predict the fine-grained human-annotated *Plausibility* scores. The best performance was achieved with our final feature set (Table 4(e)), which additionally incorporates the output from an Attentive Quality Classifier. This classifier is trained on V-JEPA 2 features to directly predict the final 'Good'/'Bad' human labels. To prevent label leakage when using the predictions from our Attentive Modules (Table 4(d,e)) as features, we employ a k-fold cross-validation strategy (we use $k = 5$). These out-of-fold predictions are then used as a "clean" feature for training our final SVM model, ensuring a fair and rigorous evaluation. This result validates that our multi-faceted feature design is highly effective at capturing the complex nuances of human judgment. The detailed classification report for this final classifier is presented in Appendix A.

## 5.3 MAIN RESULTS

**Quantitative Results.** We present our quantitative results in Table 1, evaluating our models against baselines on the held-out validation set. The evaluation uses our proposed Gemini-based scores, the temporal artifact rate, and metrics from VideoPhy2. As shown, our fine-tuned models achieve the highest average Gemini score and the best *SA* score, while also exhibiting a significantly lower temporal artifact rate. While Hunyuan attains high scores in *Plausibility* and *Continuity*, we attribute this to its tendency to generate static or moderate-action videos, which results in critically low scores for *Clarity* and *Causality*. To further validate this hypothesis, we evaluated the I2V-based models on the VBench benchmark (Huang et al., 2024) in Table 2. This analysis confirmed our observation: Hunyuan recorded the lowest *Dynamic Degree* (0.20). In contrast, our fine-tuned I2V not only improves the *Dynamic Degree* from 0.64 to 0.73 but does so without compromising other criteria. Furthermore, in direct human comparisons, our model demonstrated the best rank and the highest win rate (WR) against all baselines. Those results show the effectiveness of our synthetic data.

In addition, to validate that our model learns a generalizable understanding of physical interactions rather than overfitting to our specific data synthesis pipeline, we evaluated its performance on the PhyGenBench benchmark (Meng et al., 2024). As PhyGenBench is a text-to-video (T2V) benchmark, we adapted it for our image-to-video (I2V) setting. For each video prompt, we first employed an LLM to generate a corresponding image prompt describing the initial frame. We then utilized Flux-dev (Labs, 2024), an open-source text-to-image model, to synthesize the start image. This choice was deliberate; by using a different image generator from the one in our data creation pipeline, we could rigorously test whether our model generalizes to a novel visual distribution. As reported in Table 4, our fine-tuned model (+Fine-tuning) shows a marginal improvement over the Wan2.1 I2V baseline in the automated evaluation. However, it achieves a significant gain in human comparison, attaining the highest win rate (42%) and the best rank (1.90). We hypothesize this discrepancy stems from the limitations of the PhyGenBench automated evaluation system in the I2V context.

Firstly, when the initial image already contains complex phenomena described in the prompt (e.g., reflections in a mirror, shadows from a light source), the video model is rewarded for generating a static video, penalizing plausible motion. Sec-

Table 4: **Evaluation on the PhyGenBench. (I2V)**

| | PhyGenBench | | | | | Human Comparison | |
|---|---|---|---|---|---|---|---|
| | Mechanics | Optics | Thermal | Material | Average | Rank ($\downarrow$) | WR |
| PhyI2V | **0.51** | 0.62 | 0.54 | **0.49** | **0.55** | 2.79 | 19% |
| Hunyuan | 0.43 | 0.57 | 0.42 | 0.28 | 0.44 | 2.93 | 12% |
| Wan2.1 I2V | 0.49 | 0.59 | 0.54 | 0.39 | 0.51 | 2.38 | 27% |
| +Fine-tuning | 0.47 | **0.63** | **0.57** | 0.41 | 0.52 | **1.90** | **42%** |

ondly, the system's reliance on retrieved key-frames makes it difficult to distinguish subtle yet critical differences between videos generated from the identical start image. Notably, PhyI2V excelled in the Mechanics and Material categories. Its success in the Material likely stems from the LLM's world knowledge (e.g., vinegar is poured into a glass of litmus solution). In contrast, its high score in Mechanics appears to be an artifact of evaluation hacking; its VLM-based refinement process may overfit to the VLM-based evaluator, a bias suggested by the near-zero scores of other models' plausible videos. Nevertheless, the superior performance of our model in human evaluations demonstrates that our synthetic dataset enables robust generalization to out-of-distribution scenarios and complex physical phenomena. Qualitative results for PhyGenBench are provided in Appendix 14.

**Qualitative Results.** Figure 2 shows a qualitative comparison between baselines and our fine-tuned models. PhyI2V and Hunyuan fail to generate proper interaction and the target state change. The FLF model shows some evidence of interaction and state transition, but suffers from severe temporal artifacts; in example (a), an abrupt change occurs between subsequent frames (indicated by a red arrow) while in, (b), the video culminates in a final frame with a suddenly different appearance. The I2V model also struggles, and while it attempts the interaction, it fails to depict the target state change (e.g., not removing the grime in (b)) and introduces critical object consistency artifacts, such as a second spray can in (a) or an extra hand appearing abruptly in (b). In contrast, our zero-shot SGS produces a plausible interaction and its result. This performance is further enhanced with our fine-tuned models, as both I2V$^+$ and SGS$^+$ generate significantly clearer interactions and state transitions. Consistent with the findings in Figure 2, Figure 5 illustrates the effectiveness of SGS over FLF and I2V. While FLF introduces visual artifacts (e.g., object popping, background shifts in the red box) and I2V fails to generate any interaction, SGS successfully creates the interactions.

### 5.4 ABLATION STUDY ON SAMPLING METHODS

To validate the effectiveness of SGS, one of our core contributions, we compare human evaluation results for our initial 1,146 videos generated via six different sampling methods. As described in Section 5.1, the evaluation was based on the relative ranking of videos per prompt (where lower is better), a binary 'Good'/'Bad' label, scores for our four detailed criteria by a human annotator, and the Temporal Artifact ratio. Table 3 shows the quantitative results of our human evaluation for each sampling method. The naive approaches, FLF and I2V, exhibit a clear trade-off. While FLF performs decently on Clarity (2.96) and Causality (2.64), it suffers from a critically low Continuity score and a high Temporal Artifact rate of 0.66. In contrast, I2V excels in Continuity with a low Temporal Artifact rate of 0.13, but its scores for Clarity (2.64) and Causality (2.36), which indicate the clarity of the state transition, are lower than those of FLF. Our SGS aims to resolve this trade-off. As shown in the table, performance is quite sensitive to the initial weight of the I2V model, $\alpha$. When $\alpha \leq 0.4$, the Temporal Artifact rate remains relatively high (above 0.2). Conversely, when $\alpha$ is increased to 0.6, the Clarity and Causality scores drop sharply compared to 0.5, indicating that the guidance from the FLF model becomes insufficient. Consequently, SGS with $\alpha = 0.5$ achieves the optimal balance, attaining the highest 'Good' video ratio (0.38) and the best scores, proving to be an effective strategy that reduces artifacts while inducing plausible state transitions.

### 6 CONCLUSION

We proposed a novel, modular framework using a taxonomy-guided pipeline, visual anchors, and State-Guided Sampling to generate controllable object interaction videos. Our experiments show that fine-tuning on this synthetic data significantly enhances a model's ability to create complex interactions. While the current implementation relies on a proprietary model, the pipeline's modularity enables integration with open-source alternatives, offering a path toward transparent, reproducible data generation for robotics and creative AI.

ADDITIONAL STATEMENT

**Societal Impact.** The proposed method is based on generative video models and may inherit societal biases from the underlying base model. To mitigate potential negative impacts, we employ a standard NSFW filter to remove graphic violence and sexually explicit content.

**Reproducibility.** The source code and the dataset will be publicly available upon publication.

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

## A    SVM EVALUATOR

|  | Precision | Recall | F1-score | Support |
|---|---|---|---|---|
| Bad | 0.82 | 0.78 | 0.80 | 530 |
| Good | 0.61 | 0.65 | 0.63 | 268 |
| Macro Avg | 0.71 | 0.72 | 0.71 | 798 |
| Weighted Avg | 0.75 | 0.74 | 0.74 | 798 |
| Accuracy | | 0.74 | | |
| ROC AUC | | 0.77 | | |

Table 5: **Overall Our SVM Evaluator Performance.**

## B    TAXONOMY DETAILS

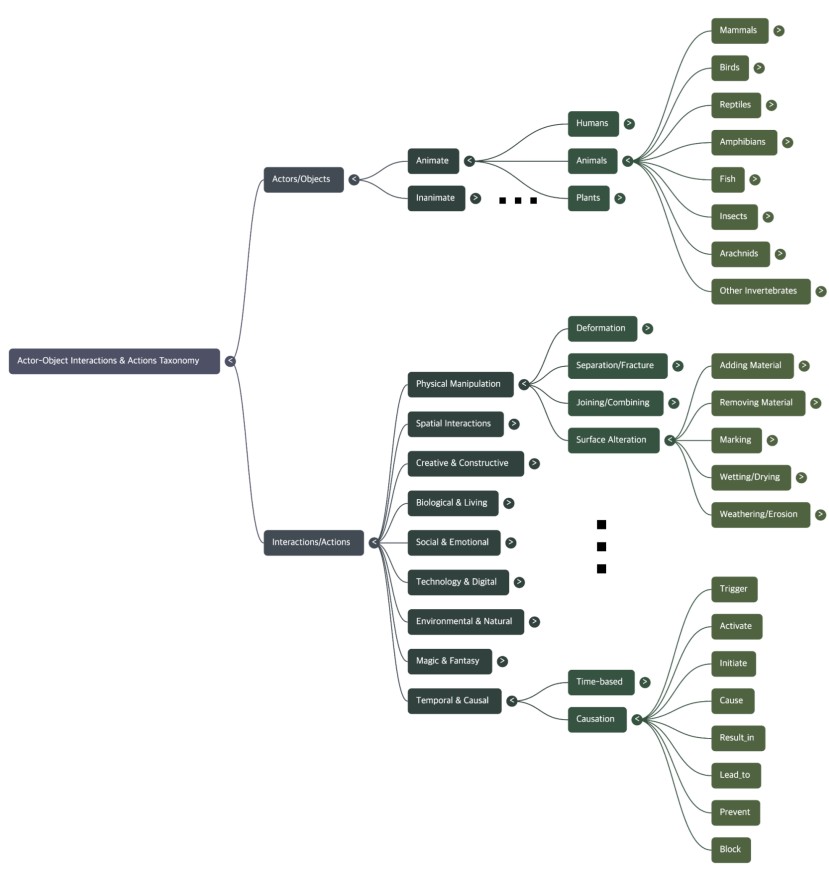

Figure 6: **The hierarchical structure of our Actor-Object and Interactions-Actions Taxonomy**, forming the basis for generating diverse interaction scenarios.

This section provides a more detailed description of the structure and content of the interaction taxonomy introduced in Section 3.1. This taxonomy is designed to enable the compositional generation of a wide and diverse range of interaction prompts, spanning from real-world interactions to creative scenarios.

The first core pillar of the taxonomy, 'Actors/Objects', defines the subjects and objects of interaction. It is broadly divided into 'Animate' and 'Inanimate' categories. The 'Animate' category includes hundreds of species of animals (including mammals, birds, insects, and dragons) and plants, while the 'Inanimate' category hierarchically organizes a wide range of objects, from everyday items like furniture, vehicles, tools, and food, to fantasy items.

The second core pillar, 'Interactions/Actions', defines the possible events. Going beyond a simple list of verbs, we have semantically grouped actions according to their physical outcomes. For example, within the 'Physical Manipulation' category, there are subgroups like 'Deformation' and 'Separation/Fracture,' which provide a structured basis for generating nuanced and specific state changes. We define approximately 500 detailed actions, including various interaction types such as spatial interactions, state changes, and creative/constructive actions.

Our taxonomy is designed around the core principles of comprehensiveness and composability. This enables our prompt generator to effectively explore a vast number of realistic and creative interaction scenarios by combining diverse elements within this structured space.

## C  GENERATION AND EVALUATION EXAMPLES

### C.1  GENERATED EXAMPLES

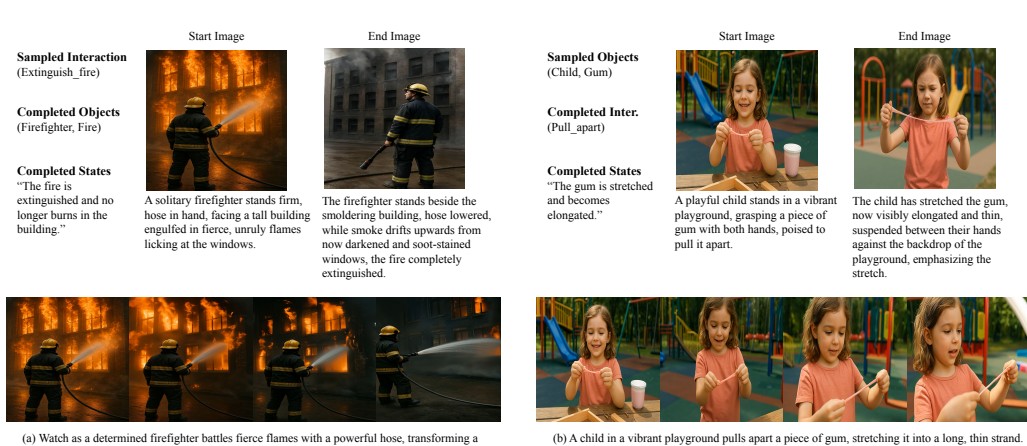

Figure 7: **An example of our data generation pipeline.** From a sampled interaction (Extinguish_fire) and objects (Child, Gum), our method completes the tuple, generates detailed prompts for the 'start' and 'end' states, synthesizes the corresponding images, and finally generates the video representing the state transition.

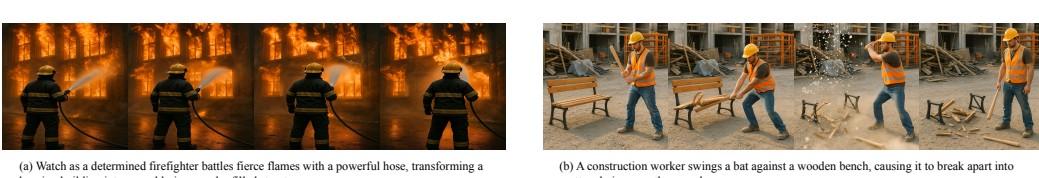

Figure 8: **Failure cases** produced by naive sampling methods. These examples illustrate the typical artifacts that our State-Guided Sampling (SGS) is designed to resolve. (a) The I2V model fails to depict the state change, resulting in a static and visually inconsistent video. (b) The FLF model creates an abrupt and unnatural transition with noticeable visual artifacts.

## C.2 VLM EVALUATION EXAMPLES

---

**Gemini Evaluation on Fig. 7 (a).**

1. Presence and Clarity of Interaction
Score: 5/5
Reason: The prompt describes a firefighter using a hose to battle flames. The video clearly and unambiguously depicts this entire process. The stream of water from the hose is visible, and its contact with the burning building is the central focus of the video, perfectly matching the prompt's intent.
2. Physical Plausibility of Video
Score: 4/5
Reason: The video is mostly physically plausible. The way the water sprays from the hose, the generation of smoke/steam as it hits the fire, and the general dimming of the scene as the flames are extinguished are all relatively natural. The fire itself looks slightly artificial, but its reaction to the water is believable.
3. Interaction State Change Causality
Score: 5/5
Reason: The prompt explicitly calls for a transformation from a "burning building" to a "smoldering, smoke-filled structure." The video perfectly illustrates this cause-and-effect relationship. It is unequivocally clear that the firefighter spraying water is the direct cause of the flames diminishing and being replaced by thick smoke.
4. Temporal Continuity and Absence of Unnatural Jumps
Score: 5/5
Reason: The video is exceptionally smooth. The movement of the firefighter, the flow of water, and the transition from fire to smoke all occur seamlessly with no noticeable jumps, flickers, or unnatural artifacts. The temporal flow is completely natural.
Total Score: 19/20

---

**Gemini Evaluation on Fig. 7 (b).**

Video Evaluation
Prompt: A child in a vibrant playground pulls apart a piece of gum, stretching it into a long, thin strand.
1. Presence and Clarity of Interaction:
Score: 5/5
Reason: The video perfectly depicts the interaction described in the prompt. A child is clearly shown holding a pink, stretchy substance (consistent with gum or slime) and pulling it apart with both hands. The entire process of stretching is the central action and is shown unambiguously.
2. Physical Plausibility of Video:
Score: 5/5
Reason: The physics of the interaction are highly plausible. The way the substance stretches, thins in the middle, and resists the pull is very consistent with the properties of a material like gum or slime. The child's movements and the static environment are all natural.
3. Interaction State Change Causality:
Score: 5/5
Reason: The cause-and-effect relationship is perfectly clear. The action of the child pulling her hands apart (the cause) directly and visibly results in the substance changing its state from a small lump to a long, thin strand (the effect). The link is immediate and undeniable.
4. Temporal Continuity and Absence of Unnatural Jumps:
Score: 5/5
Reason: The video is completely smooth and continuous. There are no noticeable jumps, flickers, or unnatural movements. The action flows naturally from beginning to end.
Total Score: 20/20
Overall Comments: This is an excellent video that perfectly matches the prompt's intent. The interaction is clear, physically believable, and demonstrates a perfect cause-and-effect relationship. The technical quality is also very high, with smooth and natural motion.

---

### Gemini Evaluation on failure case Fig. 8 (a).

Evaluation Results:

1. Presence and Clarity of Interaction: 4/5

Reasoning: The video clearly depicts the interaction specified in the prompt: a firefighter spraying a hose at a burning building. The firefighter, hose, water stream, and fire are all present and their relationship is clear. The mechanism of interaction (water hitting the fire) is mostly clear, though the exact point of impact is somewhat obscured by the flames.

2. Physical Plausibility of Video: 2/5

Reasoning: The video has several physically implausible elements. The firefighter is completely static, showing no reaction or strain from holding a high-pressure hose. The water stream itself is also static, lacking the arc and spray pattern expected from real-world physics. The fire appears to be a simple looping animation and does not react dynamically to the water being sprayed on it.

3. Interaction State Change Causality: 1/5

Reasoning: This is the video's biggest failure. The prompt explicitly asks to see the firefighter's actions ¨transforming a burning building into a smoldering, smoke-filled structure.¨However, the water being sprayed on the fire has absolutely no discernible effect. The flames do not diminish, change color, or produce more smoke. The state of the building does not change at all as a result of the interaction. The cause (spraying water) does not lead to any effect.

4. Temporal Continuity and Absence of Unnatural Jumps: 5/5

Reasoning: The video is perfectly smooth. It functions like a cinemagraph where certain elements (the fire) are animated in a loop while others remain static. There are no jumps, glitches, or unnatural warping, resulting in a very stable temporal flow.

Total Score: 12/20

Overall Comments: The video successfully establishes the scene and the intended interaction from the prompt. However, it completely fails to show the consequence of that interaction. It's more of a "living image" than a video depicting a cause-and-effect event. While temporally smooth, the lack of physical reaction and state change makes the core interaction feel pointless and unrealistic.

---

### Gemini Evaluation on failure case Fig. 8 (b).

Video Evaluation

1. Presence and Clarity of Interaction: 2/5

Reasoning: The prompt specifies the worker "swings a bat against a wooden bench." The video shows the worker preparing to swing (0:00) and the immediate aftermath of the bench breaking (0:01), but the critical moment of the bat making contact with the bench is completely omitted. The interaction is implied by the result, but the mechanism of the interaction is not depicted, making the prompt's intent poorly reflected.

2. Physical Plausibility of Video: 1/5

Reasoning: The video has several major physical implausibilities. The bench doesn't break as if hit by a bat; it explodes outwards from its center as if a charge was detonated within it. Most significantly, a second, identical baseball bat appears on the ground from 0:02 onwards, while the worker is still holding the original bat. This is physically impossible.

3. Interaction State Change Causality: 2/5

Reasoning: While the bench does break (a state change), the causal link is very weak. Because the impact is not shown, the bench's destruction feels more like a sudden, spontaneous event that happens to coincide with the worker's action rather than being a direct result of it. The "cause" (the hit) is missing, making the "effect" (the break) feel disconnected.

4. Temporal Continuity and Absence of Unnatural Jumps: 1/5

Reasoning: The video suffers from severe temporal discontinuity. There is a major jump cut between 0:00 and 0:01, skipping the entire swing and impact action. Furthermore, an object (a second bat) teleports into the scene at 0:02, which is a drastic and unnatural jump in the video's state.

Total Score: 6/20

---

In designing our VLM-based evaluation pipeline, we initially experimented with using a structured JSON format to query the model and receive its scores. However, we empirically found that forcing the model to adhere to a rigid JSON schema significantly degraded its evaluation performance compared to using an unconstrained natural language prompt. Consequently, we adopted a more effective two-step process: we first prompt the model using natural language to elicit a detailed, free-form text evaluation, and then parse this natural language output to extract the final structured scores.

## D  TEMPORAL ARTIFACT DETECTOR

| Correlation | Accuracy | Precision | Recall | F1-score |
|---|---|---|---|---|
| 0.637 | 0.865 | 0.572 | 0.820 | 0.674 |

Table 6: **Performance of the temporal artifact detector.** The Pearson Correlation is calculated against the raw (1-5) human-rated Continuity scores. The binary classification metrics are based on a threshold where a human *Continuity* score $\leq 2.0$ defines the positive class.

To validate the performance of our proposed temporal artifact detector, we used the 1,146 human-annotated videos as ground truth. The detector's continuous prediction score showed a high Pearson correlation of 0.64 with the 1-5 human-rated Continuity scores.

Furthermore, we measured its binary classification performance on detecting videos with severe artifacts. We define the positive class ('artifact present') as videos with a human-rated Continuity score of 2.0 or lower. The classification performance is shown in Table 6. As seen in the table, our detector achieves a high Recall of 0.82, successfully identifying the majority of videos that contain actual artifacts. While its Precision of 0.57 indicates the presence of some false positives, the overall F1-score of 0.67 demonstrates reliable performance. The high Recall aligns with our primary goal for dataset filtering, where correctly identifying poor-quality samples is paramount. This result validates that the detector is effective enough to serve as an auxiliary feature in our evaluation framework.

## E  ADDITIONAL QUALITATIVE RESULTS

Figure 9 provides a clear visual demonstration of the 'ghosting effect,' a critical artifact that arises from naive score mixing approaches, and illustrates how our State-Guided Sampling (SGS) method resolves it. The examples labeled 'Constant Interpolation' show the result of using a fixed, linear weight to combine the FLF and I2V models. This method's rigid adherence to the target end-frame forces an unnatural, translucent overlay of the start and end states, which is particularly prominent in the final frames of the sequence.

As shown in the right column, SGS effectively mitigates this artifact. By exponentially decaying the FLF model's influence towards the end of the sequence, SGS allows the I2V model's strength in maintaining local coherence to dominate. This results in a physically plausible and temporally consistent final state that naturally evolves from the preceding motion, highlighting the necessity of our dynamic weighting scheme.

In addition to this analysis, we provide further qualitative results, including direct comparisons with baseline models and additional examples contrasting the FLF, I2V, and SGS sampling methods.

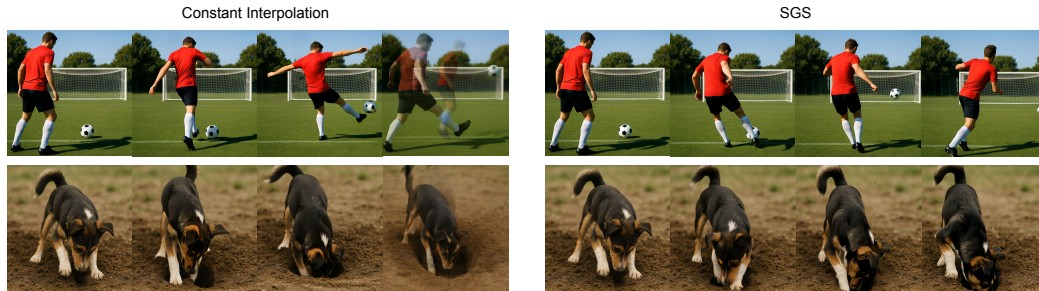

Figure 9: **Qualitative examples of the 'ghosting effect'.** Naive Constant Interpolation (left column) results in an unnatural, translucent overlay in the final frames. In contrast, our proposed SGS (right column) resolves this artifact by dynamically adjusting model influence, producing a clear and temporally coherent final state.

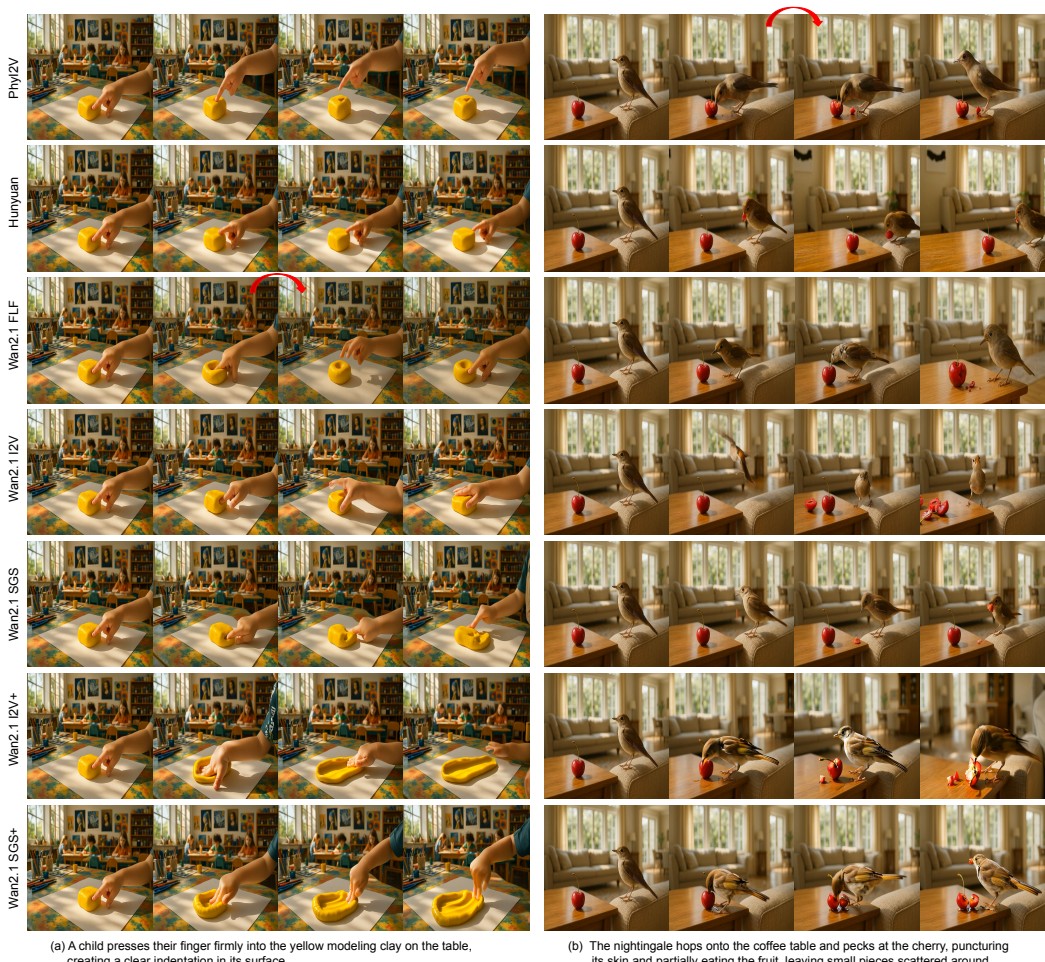

(a) A child presses their finger firmly into the yellow modeling clay on the table, creating a clear indentation in its surface.

(b) The nightingale hops onto the coffee table and pecks at the cherry, puncturing its skin and partially eating the fruit, leaving small pieces scattered around.

Figure 10: **Qualitative Baseline Comparison.**

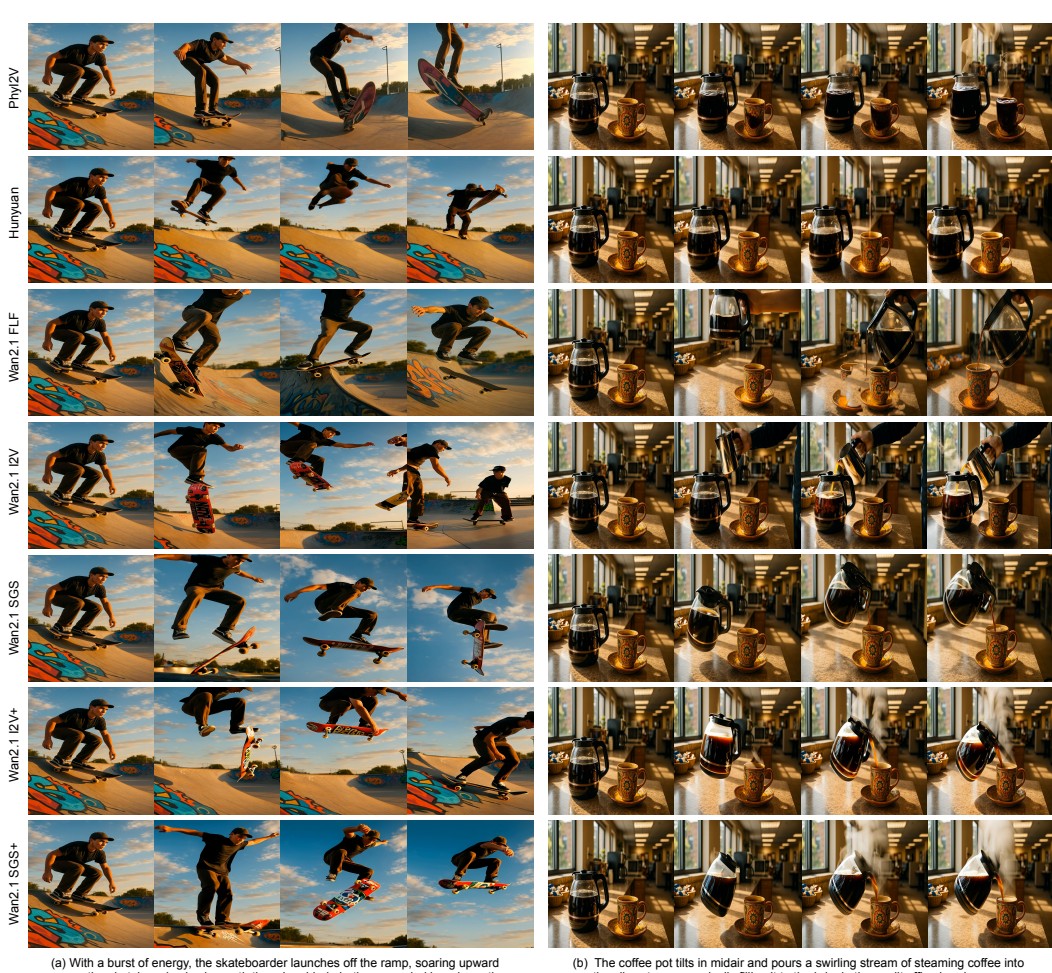

(a) With a burst of energy, the skateboarder launches off the ramp, soaring upward as the skateboard spins beneath them in mid-air, both suspended in a dramatic, sunlit leap.

(b) The coffee pot tilts in midair and pours a swirling stream of steaming coffee into the vibrant mug, magically filling it to the brim in the sunlit office break room.

Figure 11: **Qualitative Baseline Comparison.**

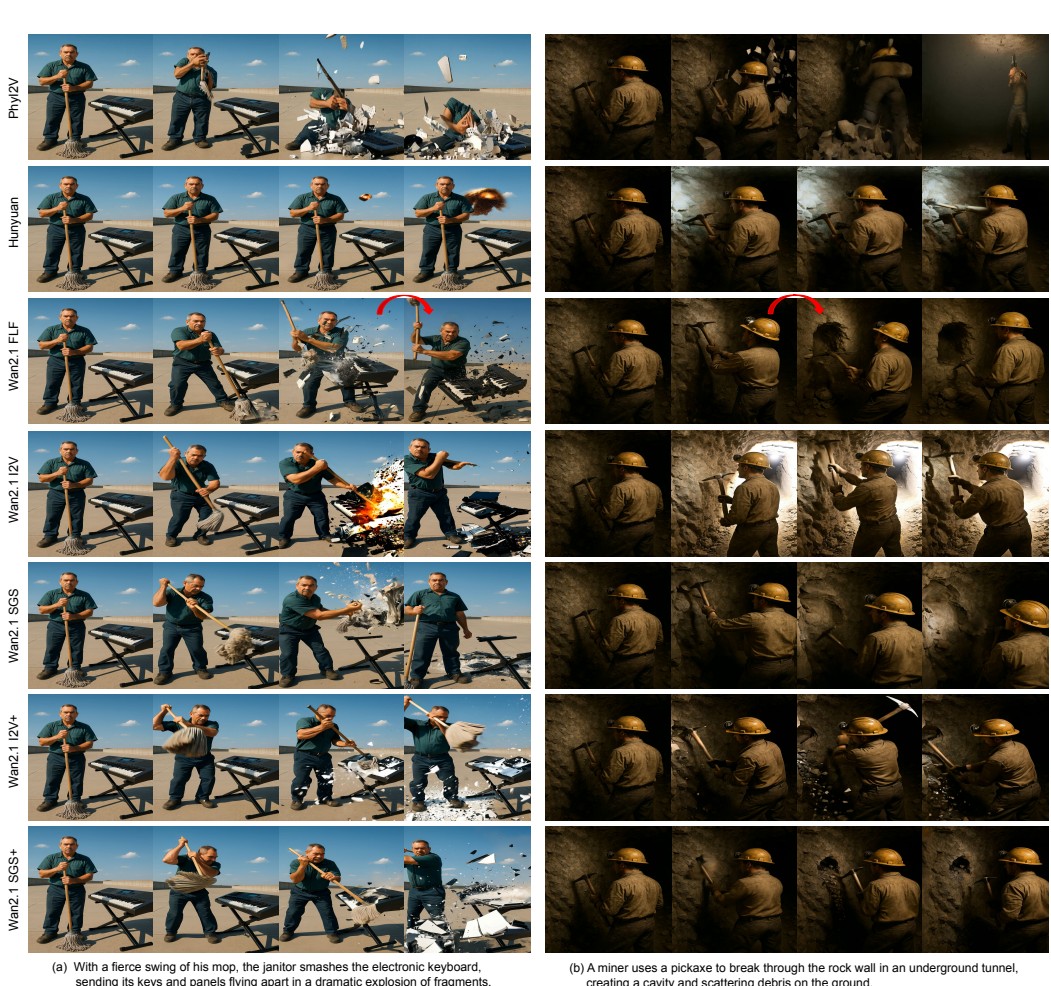

(a) With a fierce swing of his mop, the janitor smashes the electronic keyboard, sending its keys and panels flying apart in a dramatic explosion of fragments.

(b) A miner uses a pickaxe to break through the rock wall in an underground tunnel, creating a cavity and scattering debris on the ground.

Figure 12: **Qualitative Baseline Comparison.**

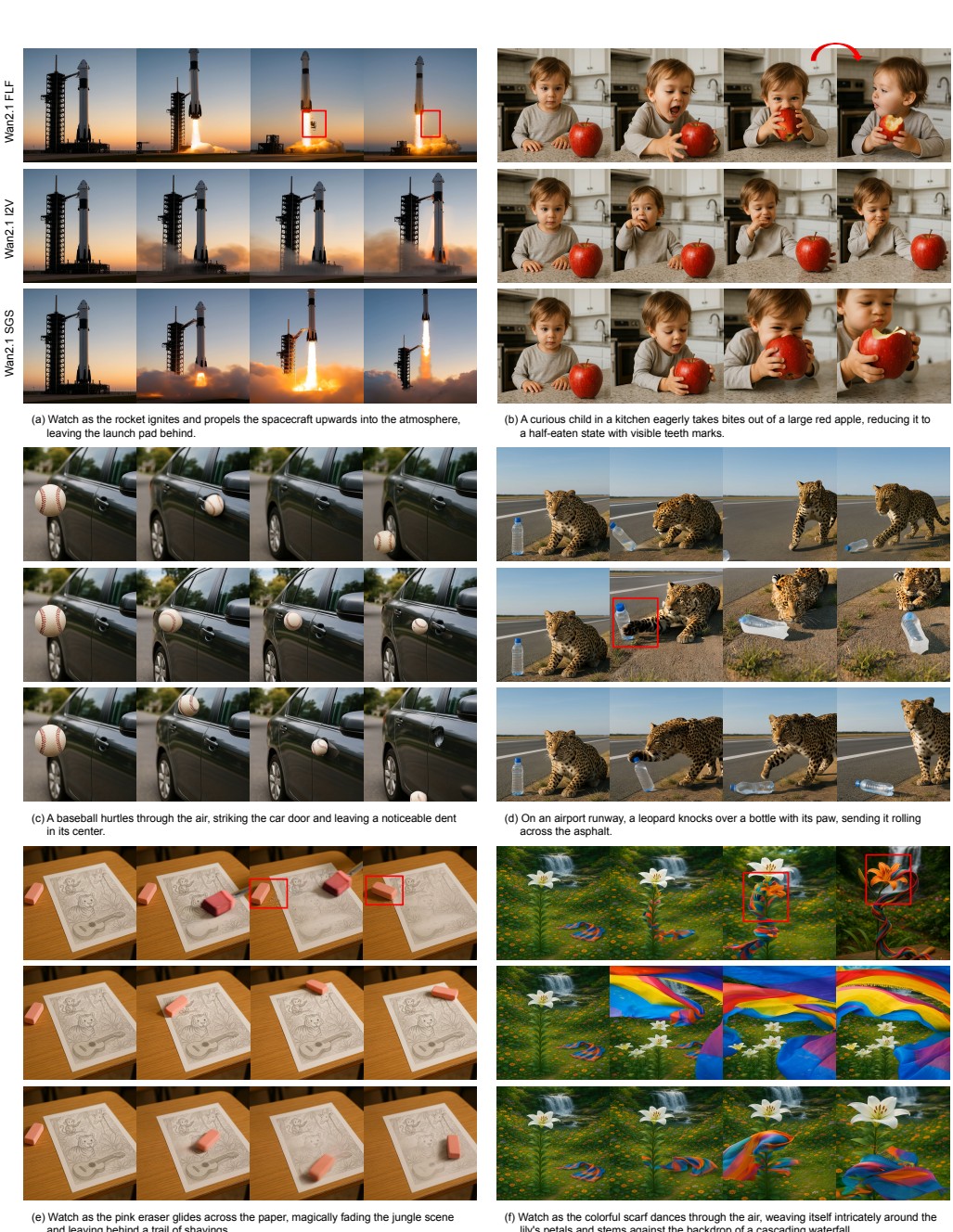

Figure 13: **Qualitative comparison of SGS against naive sampling methods.**

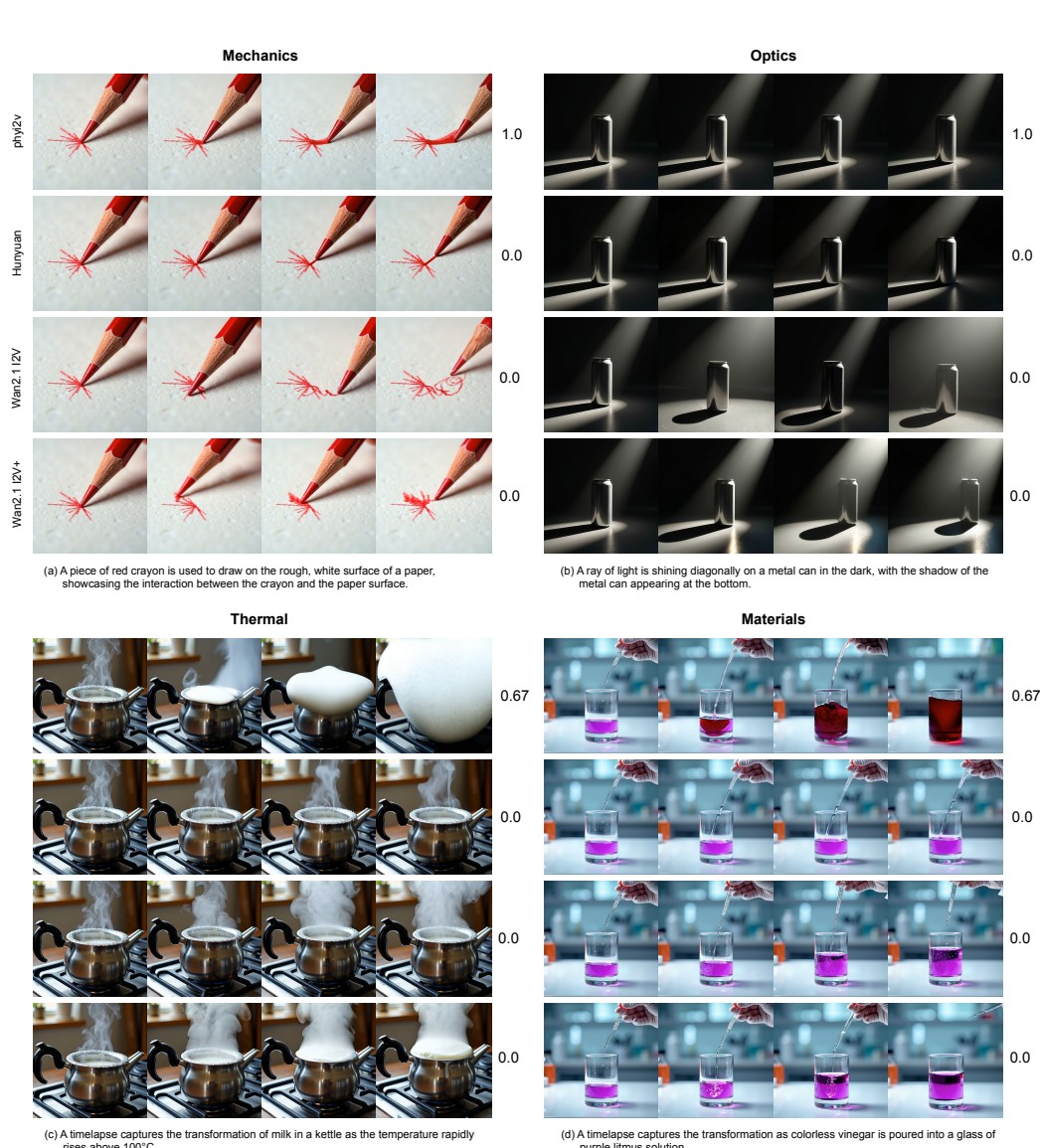

Figure 14: **Qualitative comparison on PhyGenBench.** While our model produces plausible videos in (a-c), only PhyI2V receives high scores, suggesting a scoring bias where its VLM-based refinement games the VLM-based evaluator. In contrast, PhyI2V's high score in (d) stems from a valid advantage in world knowledge (litmus solution chemistry).

## F  SYSTEM PROMPTS

### F.1  TAXONOMY GUIDED PROMPT GENERATOR

```
Context:
- Object 1: '<Object1_Name>' (Category: <Object1_Taxonomy_Path>)
- Object 2: '<Object2_Name>' (Category: <Object2_Taxonomy_Path>)

Task:
First, evaluate if a meaningful interaction between Object 1 and
    Object 2 is likely in this scene. The objects were chosen randomly
    , so they might not make sense together.

If a meaningful interaction exists:
1. Suggest *one* such single interaction verb or short phrase (like '
    Cut', 'Pour', 'Collide_with', 'Stack_on') describing how Object 1
    might interact with Object 2 in this scene.
2. Describe the primary **major state change** resulting from this
    interaction, ensuring a significant alteration in the shape or
    condition of at least one object (e.g., 'Object 2 shatters', '
    Object 1 melts onto Object 2', 'Object 2 is torn apart'). Focus on
     a visually impactful consequence.

Respond ONLY in the strict format: Interaction: [Interaction Name],
    StateChange: [Description of state change]
Example: Interaction: Dig, StateChange: A hole appears in the ground
If no meaningful interaction seems likely, respond exactly with 'None
    '.
```

Prompt 1: Object-Centric Tuple Generation.

```
Context:
- Interaction: '<Interaction_Name>' (Category: <
    Interaction_Taxonomy_Path>)

**IMPORTANT**: The following combinations have been previously
    generated for this interaction:
- Object1: <Prev_O1_Name>, Object2: <Prev_O2_Name>, StateChange: <
    Prev_StateChange_Desc>
- ... (more examples if they exist) ...

You MUST suggest a completely different combination. Avoid repeating
    any of the above Object1, Object2, Scene combinations. Think of
    alternative scenarios, different object types, or different
    settings where this interaction could occur.

Task: Suggest a creative and plausible scenario for the given
    interaction.
1. Suggest an Object 1 (typically the actor or initiator).
2. Suggest a distinct Object 2 (typically acted upon or involved).
3. Describe the primary **state change** resulting from this
    interaction. This description **must** indicate a clear and
    definite change in the shape, condition, or relative position of
    at least one object (e.g., 'Object2 is broken into two pieces', 'A
     hole appears in Object2'). Focus on the most direct, observable
    consequence.

Respond ONLY in the strict format: Object1: [Name], Object2: [Name],
    Scene: [Name], StateChange: [Description of state change]
Example response: Object1: Dog, Object2: Ground, Scene: Backyard,
    StateChange: A hole appears in the ground
```

Prompt 2: Interaction-Centric Tuple Generation.

```
1  You are an expert assistant. Your primary goal is to generate two
       visually rich and detailed image prompts based on the provided
       scenario details: 'Prompt_Before_Interaction' and '
       Prompt_After_Interaction'. The prompts should describe a realistic
        and well-composed scene.
2
3  For both prompts:
4  - Use descriptive, evocative language. Focus on creating a complete,
       believable scene.
5  - Describe subject(s), their state, their relationship to the
       environment.
6  - Avoid any conversational filler, questions, or explanations *in your
        generated prompts*.
7  - **IMPORTANT: Avoid negative terms like "not", "no", "without", "
       absent", "missing", "lack of", etc. Image generation models
       struggle with negative concepts. Instead, describe what IS present
        and visible.**
8
9  'Prompt_Before_Interaction': Describe a complete, static scene, paying
        close attention to realistic object placement and a coherent
       environment.
10 'Prompt_After_Interaction': Describe the scene *after* the interaction
        for an image editing model. Clearly depict the final state,
       ensuring the 'Expected State Change' is visually and dramatically
       represented.
11
12 Adhere strictly to the detailed scenario context, plausibility mode
       instructions, and JSON output format that will be provided in the
       subsequent parts of the full instruction set you receive.
13
14 ---
15 Scenario Context:
16 - Object 1 (Actor/Initiator): <Object1_Name> (<Object1_Taxonomy_Path>)
17 - Object 2 (Acted Upon): <Object2_Name> (<Object2_Taxonomy_Path>)
18 - Interaction: <Interaction_Name> (<Interaction_Taxonomy_Path>)
19 - Expected State Change: <StateChange_Description>
20
21 - Important: If multiple instances of Object 1 or Object 2 are
       involved in the interaction, explicitly specify the number of
       objects in both before and after prompts.
22
23 Generate two detailed prompts based on the scenario, focusing on *
       physical plausibility and realism*:
24
25 1.  **Prompt_Before_Interaction:** A detailed prompt for generating a
       static, photorealistic image. Describe a complete scene with
       realistic lighting, shadows, and composition. **Crucially, all
       objects must be realistically placed within the scene (e.g., on a
       surface, held by a person). Objects must NOT be floating,
       levitating, or positioned in a physically impossible way.**
       Describe the background and the spatial relationship between
       objects to create a believable and interesting context.
26
27 2.  **Prompt_After_Interaction:** A detailed instruction prompt for an
        *image editing* model. Describe the physically plausible result
       of the interaction. Clearly state the final positions and
       conditions of the objects, **and ensuring the Expected State
       Change is visually represented in a compelling way**. Focus on
       realistic changes to the scene. Use positive descriptive language
       - describe the new state rather than what is no longer present.
28
29 If human involvement is typically required for this interaction, imply
        or explicitly describe the necessary human action to make the
       scene realistic.
```

```
30
31 Output the results in JSON format , exactly like this :
32 {
33    "Prompt_Before_Interaction": "[brief realistic before image prompt
         ]",
34    "Prompt_After_Interaction": "[brief realistic after edit instruction
          incorporating state change]"
35 }
```

Prompt 3: Physically Plausible Start and End Prompt Generation.

```
1 Generate two creative and visually rich prompts based on the scenario ,
        suitable for a *cinematic or animated context*:
2
3 1.  **Prompt_Before_Interaction:** Describe a visually interesting
        scene *just before* the interaction . Emphasize dynamic composition
        . Objects can be anthropomorphized or positioned unusually for
        storytelling effect , but the scene should still be artistically
        coherent . Use positive descriptive language .
4
5 2.  **Prompt_After_Interaction:** Describe the scene *after* the
        interaction for an *image editing* prompt . Clearly state the final
         positions of the objects , showing the result of the interaction
        in a visually striking or story-driven way , **and ensuring the
        Expected State Change is represented**. The interaction might be
        autonomous or stylized . Focus on visual storytelling . Use positive
         descriptive language .
6
7 Human involvement is optional; if absent , describe the objects acting
        with clear intent or purpose .
8 }}
```

Prompt 4: Cinematic and creative Start and End Prompt Generation. Replace L23-30 in Prompt 3.

## F.2  VIDEO PROMPT GENERATOR

```
1 Your task is to generate a single , concise video prompt OR filter the
        sample if the input is invalid .
2
3 Input Analysis :
4 1. Critically evaluate the provided 'before' and 'after' images , the
        scenario details , and the image generation prompts .
5 2. Check for:
6     - Consistency: Do the images reasonably match the scenario details
         (objects , interaction , state change) and the image prompts?
7     - Logical Transition: Does the change from the 'before' to the '
        after' image plausibly represent the specified interaction and
        result in the **Intended State Change (<StateChange_Description>)
        **?
8     - Image Quality/Clarity: Are the images clear enough to understand
         the scene and the interaction?
9
10 Filtering Condition :
11 - If you determine that the input is inconsistent , the images are too
        low quality/unclear , the described interaction doesn't match the
        visual change (especially the specified state change (<
        StateChange_Description>)), or the scenario is nonsensical based
        on the provided images and text , respond *only* with the exact
        string: 'FILTER_SAMPLE'
12
13 Video Prompt Generation (if input is valid):
14 - If the input passes your evaluation , generate a single , concise , and
         motion centric video prompt (single sentence).
```

```
15 - This prompt must describe the dynamic action (<Interaction_Name>)
     that transforms the 'before' image into the 'after' image,
     resulting in the **Intended State Change (<StateChange_Description
     >)**.
16 - Focus on the action and the resulting state change.
17 - Maintain consistency with the visual style, objects, and scene
     depicted in the images.
18 - Ensure the prompt reflects the requested '<Plausibility_Mode>'
     plausibility (physical realism or cinematic flair).
19 - **Decision Making:** You must decide whether the interaction
     requires human intervention based on the interaction type and
     plausibility mode, then adjust your prompt accordingly.
20
21 **Examples of Human Intervention Decision:**
22 - Physical Mode: "Bat hits window" -> "A person grabs the baseball bat
     and swings it forcefully, shattering the window glass"
23 - Cinematic Mode: "Bat hits window" -> "The baseball bat levitates and
     swings autonomously, magically shattering the window"
24 - Physical Mode: "Ball rolls down hill" -> "A ball rolls down the
     grassy hill" (no human needed)
25 - Physical Mode: "Knife cuts bread" -> "A person picks up the knife
     and carefully slices through the bread loaf"
26 - Cinematic Mode: "Knife cuts bread" -> "The knife glides through the
     air and slices the bread by itself"
27
28 - Output *only* the generated video prompt as a single string, without
     any introductory text, labels, or explanations.
29
30 Scenario Details:
31 - Object 1: <Object1_Name> (<Object1_Taxonomy_Path>)
32 - Object 2: <Object2_Name> (<Object2_Taxonomy_Path>)
33 - Interaction: <Interaction_Name> (<Interaction_Taxonomy_Path>)
34 - Intended State Change: **<StateChange_Description>**
35 - Plausibility Mode: <Plausibility_Mode>
36
37 Image Generation Prompts Used:
38 - Before Image Prompt: <Before_Prompt_Text>
39 - After Image Prompt: <After_Prompt_Text>
40
41 Here are the images:
```

Prompt 5: Video Prompt Generation.

## F.3 VIDEO EVALUATOR

```
1 Generated Video Evaluation Guidelines (Object Interaction Focused -
    Score Based & Prompt Considered)
2 Objective: These guidelines are designed to quantitatively evaluate
    the quality of object interactions within generated videos. The
    focus is on assessing how naturally and realistically the video
    portrays interactions, the smoothness of temporal flow, and how
    well it reflects the intent of the provided text prompt.
3
4 Evaluator: You will watch the generated video, review the accompanying
    text prompt, and conduct the evaluation based on the criteria and
    scoring scale below.
5
6 Evaluation Criteria and Scoring Scale:
7
8 Please watch each video and assign a score from 1 to 5 for each of the
    following three criteria. When evaluating, consider the content
    of the text prompt used for video generation to assess the intent
    and implementation of the interaction.
9
```

```
10  1. Presence and Clarity of Interaction (1-5 points):
11  * (Prompt Consideration): If the prompt specified a particular
       interaction (e.g., "A hits B," "A pushes B," "A and B collide"),
       is this specific interaction visibly depicted in the video? Is the
        mechanism of interaction (the contact, the force transfer)
       clearly represented according to the prompt's intent, rather than
       just implied by the outcome?
12  * 1 point: No interaction between the specified objects is depicted,
       or only negligible, unrelated movements occur. The prompt's
       interactive intent is entirely absent.
13  * 2 points: Interaction is attempted or implied as per the prompt, but
        the critical moment of interaction (e.g., contact, force transfer
       ) is missing, glossed over, or fundamentally flawed (e.g., objects
        appear to affect each other without clear contact, effects are
       misaligned with supposed actions, or objects pass through each
       other when contact is expected). The outcome might occur (e.g., an
        object falls), but the mechanism described in the prompt is not
       actually depicted, making the intent poorly reflected.
14  * 3 points: Interaction is depicted, and the general intent is
       understandable. However, key aspects of the interaction process (
       the 'how') are unclear, briefly obscured, or slightly misaligned.
       For instance, contact might occur, but it's too quick to properly
       assess, or partially hidden in a way that makes the exact nature
       of the engagement ambiguous. Prompt intent is partially reflected.
15  * 4 points: Clear interaction is depicted. It's relatively easy to
       understand which objects are interacting and how they are
       physically engaging (e.g., contact is visible and plausible). The
       mechanism of interaction is mostly clear, and the prompt intent is
        mostly well reflected.
16  * 5 points: The complete process of interaction between two or more
       objects, as specified or implied by the prompt, is very clearly
       and unambiguously depicted. The mechanism of interaction (e.g.,
       pushing, pulling, colliding, contact points) is visually explicit,
        sustained enough to be observed, and entirely consistent with the
        prompt's intent.
17
18  2. Physical Plausibility of Video (1-5 points):
19  * (Prompt Consideration): By default, judge only how well the video
       obeys everyday physical laws-ignore any prompt or intended effects
       .
20  * (Exception): If the prompt explicitly calls for non-standard or "
       magical" physics, then judge how consistently the video realizes
       that specified "magic" or special effect.
21  * 1 point: The video completely defies physical laws and is highly
       unnatural (e.g., ignoring gravity, objects passing through each
       other, unrealistic deformations).
22  * 2 points: Many physically awkward aspects. The sense of weight,
       material properties, etc., of the objects is barely noticeable.
23  * 3 points: Some physically awkward parts exist, but overall it doesn'
       t significantly deviate from common sense. Basic collision,
       movement, etc., are implemented.
24  * 4 points: The video appears mostly physically plausible. Object
       movements, velocity changes, etc., are relatively natural. Minor
       awkwardness might be present.
25  * 5 points: The video aligns very well with real-world physics.
       Gravity, friction, reaction upon collision, object mass/material
       properties appear naturally reflected.
26
27  3. Interaction State Change Causality (1-5 points):
28  * (Prompt Consideration): Does the video clearly demonstrate that the
       change in an object's state (e.g., B moving, breaking, changing
       color, etc.) is a direct and understandable result of the
       interaction with A, as described or implied by the prompt? Does
       the prompt specify a particular resulting state change, and is
       this causal link evident?
```

```
29  * 1 point: The state change of the interacted object appears random,
        spontaneous, or completely unrelated to the depicted interaction.
        The prompt's implied or stated consequence of the interaction is
        not causally linked to the interaction itself.
30  * 2 points: A state change occurs in the interacted object, but the
        causal link to the interaction is very weak, highly ambiguous, or
        seems to be coincidental rather than a direct result. The prompt's
        intended outcome feels disconnected from the interaction shown.
31  * 3 points: The interaction leads to a state change in the object, but
        the causality is not entirely clear or immediate. There might be
        other distracting elements, or the exact moment/reason for the
        state change is somewhat obscure, making the link to the prompt's
        intended consequence partially unclear.
32  * 4 points: The change in the object's state is clearly and directly
        caused by the interaction. The "before and after" states are
        distinct, and the interaction serves as a convincing trigger,
        mostly aligning with the prompt's implied or explicit causal chain
        .
33  * 5 points: The video perfectly illustrates the cause-and-effect
        relationship. The interaction unequivocally and visibly leads to
        the specific change in the object's state as described or
        logically implied by the prompt. The sequence of interaction
        leading to the resultant state change is unambiguous and
        compelling.
34
35  4. Temporal Continuity and Absence of Unnatural Jumps (1-5 points):
36  * (Prompt Consideration): Unless the prompt intentionally requested
        scene transitions or effects, is the temporal flow natural within
        the depicted segments?
37  * 1 point: Severe issues throughout the video, such as frame drops,
        scene jumps that disrupt the depicted action, objects teleporting
        or drastically changing shape unnaturally.
38  * 2 points: Unnatural jumps between frames or abnormal object
        movements (e.g., flickering, warping unrelated to interaction) are
        frequently noticeable, disrupting the viewing flow of the
        depicted scenes.
39  * 3 points: Minor jumps or unnatural movements appear intermittently,
        but they don't significantly hinder understanding the flow of the
        action that is shown.
40  * 4 points: The temporal flow of the depicted action is generally
        smooth with almost no jumps. Object movements are continuous.
41  * 5 points: Video playback is very smooth, and the temporal flow from
        the beginning to the end of the depicted interaction is completely
        natural. No frame drops or abnormal object movements are observed
        .
42
43  Evaluation Method:
44
45  First, review the text prompt provided with each video.
46
47  Watch the video and assign a score between 1 and 5 for each of the
        three criteria (1. Presence/Clarity of Interaction, 2. Physical
        Plausibility, 3. Interaction State Change Causality, 4. Temporal
        Continuity). Evaluate by considering the prompt content.
48
49  Briefly noting the reason for each score or referencing specific video
        segments (timestamps) and their relevance to the prompt content
        helps improve evaluation reliability. (e.g., "Interaction Clarity
        3 points - Prompt requested 'ball knocking over a cup', but the
        ball just passes by the cup (0:07s).")
50
51  Reporting Results:
52
53  Record the scores for the four criteria for each video in the provided
        format (e.g., spreadsheet, evaluation document).
```

```
54
55 Sum the scores for each criterion to calculate the Total Score (
        minimum 4 points ~ maximum 20 points) and record it as well.
56
57 If necessary, you can add brief comments on the video's implementation
        level relative to the prompt and the overall quality of the
        interaction alongside the total score.
58
59 We hope these guidelines facilitate consistent and quantitative video
        evaluations. Thank you for your participation in the evaluation.
60
61 ---
62 Please read this instruction. After this, I will provide the video and
        its prompt.
```

Prompt 6: Gemini video evaluation prompt.

## G   FUTURE WORK

Our research will focus on two key areas: leveraging a more robust understanding of physics in the model and broadening the framework's practical applications. To expand the model's current implicit knowledge, we plan to directly integrate simplified physics engines. These engines will guide the video generation process, ensuring that physical laws are respected from the beginning rather than merely filtering out implausible results. Besides, we plan to expand our taxonomy beyond its current 1,300 objects and 500 interactions. Generating a much larger dataset than our final training set of 1,525 videos will allow us to employ more advanced preference tuning techniques, building on our initial use of human-labeled data and enabling us to better capture the nuances of realistic interactions.

For this work to have a broader impact, it must be both accessible and applicable. Therefore, we will utilize the open-source image editors and VLMs. This will address the current limitation of relying on a single third-party model. An open framework can accelerate progress in critical areas such as robotics, where a core goal is to equip world models with a robust understanding of physical cause and effect. In addition to robotics, these advancements will enable the creation of dynamic and interactive content for virtual reality and other immersive 4D applications.

