# OpenReview forum: "Zero-to-Interaction: Generating Dynamic Videos from Synthetic State Transitions"
_ICLR.cc/2026/Conference — ICLR 2026 Conference Withdrawn Submission_

### Official Review · Reviewer_c29K · 2025-10-29

**Soundness:** 2
**Presentation:** 2
**Contribution:** 1
**Rating:** 4
**Confidence:** 3

**Summary:**

This paper proposes a pipeline for generating synthetic interaction videos, which leverages a structured taxonomy to create prompts, generates "start" and "end" state images, and uses State-Guided Sampling to produce seamless videos. It also develops a benchmark to evaluate such videos, assessing four key criteria via a hybrid framework integrating VLM scores and auxiliary features to align with human judgments.

**Strengths:**

From some results, it can be seen that the quality of interactions in the generated videos has indeed improved.

**Weaknesses:**

- Collecting better data to tune the model for improved performance seems obvious; this aspect lacks academic innovation and is more of an engineering effort.
- The multi-stage method for generating video data with high-quality interactions is very straightforward. It is a common approach used in both industry and academia to construct large-scale image/video editing datasets, making it hard to perceive impactful insights.
- The authors discuss the respective advantages and disadvantages of the I2V model and FLF model, then attempt to achieve optimal results using SGS. From the results provided by the authors, I do not actually perceive obvious differences between the methods. Another question is whether the poor performance of the FLF model is simply due to inadequate training of Wan-FLF; a better FLF model might eliminate the need for methods like SGS, which undermines the contribution and innovation of this approach. (to my knowledge, the FLF model is not a universally accepted term in academia, and the authors have not provided references for it)

**Questions:**

Can the quality of interactions in videos generated by the model be improved by repeatedly trying different text prompts and image prompts? Can this method be included in the comparative experiments?

---

### Official Review · Reviewer_FK5a · 2025-11-01

**Soundness:** 2
**Presentation:** 2
**Contribution:** 1
**Rating:** 2
**Confidence:** 4

**Summary:**

This paper proposes a framework for adapting the pre-trained video generative model to follow the dynamic rules and synthesize physically plausible transitions. The proposed framework consists of prompt editing, image synthesis, and state-guided sampling. Its performance is evaluated on several tasks across diverse dimensions.

**Strengths:**

- The paper is well-motivated, and the proposed method is sound. This paper proposes an entire framework to enhance the dynamic following ability of pre-trained video generative models.
- This paper also includes a detailed evaluation pipeline to test whether the generated videos are temporally and dynamically consistent.

**Weaknesses:**

- The proposed method is not novel, and there are limited insights in this paper. The start-and-end-frame condition is not new. And the proposed state-and-guided sampling is not theoretically sound. I suspect whether the weighted velocity preserves a valid marginal data distribution $p(x_0)$. Any theoretical analysis is welcome to be added to the paper.
- Moreover, the proposed method requires additional computation for generating a video. For example, the prompt and image generation that needs a large model (e.g., GPT4o) and a double denoising forward process.
- The empirical performance is not obviously advantageous compared with baselines. For example, the results in Table 1 and Table 4. Additionally, no standard deviation is provided.
- The demos provided in the paper and webpage are not that impressive. Considering the more computation required by the method, I expect to see more distinct advantages.
- Overall, while I appreciate the engineering efforts made in this work, I tend to a 'reject' rating considering the limited novelty and insights.

**Questions:**

- Does the proposed method work for other pre-trained video models? Can this method be adapted to an autoregressive transformer-based architecture?
- Are there any failure cases? I believe the video generation quality still heavily depends on the performance bound of the pre-trained models.

---

### Official Review · Reviewer_91ac · 2025-11-01

**Soundness:** 3
**Presentation:** 3
**Contribution:** 3
**Rating:** 4
**Confidence:** 4

**Summary:**

This paper introduces Zero-to-Interaction, a modular framework for generating physically consistent object-interaction videos from synthetic data. It leverages large vision-language models to construct structured prompts and generate paired “start” and “end” state images, which serve as anchors for interaction synthesis. A new **State-Guided Sampling (SGS)** method then blends I2V and FLF (start-and-end-frame conditioned) velocity fields to produce smooth transitions. An automated **VLM-based evaluation framework** further assesses interaction quality. Experiments show clear gains over baselines such as HunyuanVideo, PhyI2V, and Wan 2.1 in both quantitative and human evaluations.

**Strengths:**

### Originality
1. The framework of this article is innovative, using VLM as a proxy for preprocessing the input, the idea of starting and ending states is original, and the proposed State-Guided Sampling algorithm is also original.

### Technical Quality
1. The taxonomy-based prompt system is well-structured and scalable, yielding diverse and creative interaction scenarios beyond existing datasets.
2. The hybrid VLM + auxiliary metric system (plausibility probe, artifact detector, etc.) is a notable contribution for automated video assessment, aligning well with human judgment.

### Clarity
1. The implementation is well-documented and modular, with commitment to releasing data, code, and evaluation tools, enhancing reproducibility and community impact.

### Significance
1. The paper tackles a relatively underexplored problem: enabling video generation models to represent physically consistent object interactions and state transitionsKey components (e.g., GPT-4o, Gemini-2.5) and commercial image editors reduce reproducibility and limit transparency, even though the pipeline itself is modular., a capability that holds significant relevance for fields such as robotics, world modeling, and visual simulation.

**Weaknesses:**

1. Key components (e.g., GPT-4o, Gemini-2.5) and commercial image editors reduce reproducibility and limit transparency, while also increasing the overall computational and financial cost of using the proposed framework.
2. Since the evaluation system partially reuses VLM-based components similar to those in training or data generation, it may not fully avoid self-consistency bias. Independent benchmarks could strengthen credibility.
3. While SGS improves temporal smoothness, it introduces two separate models, which increase the overall system complexity and computational cost; however, the paper lacks corresponding analysis or justification for this design choice.

**Questions:**

1. What is the practical scalability of the proposed pipeline — specifically, how long and how costly is it to generate one high-quality interaction video when the entire pipeline is used (including VLM-based preprocessing for prompt construction and for synthesizing the start/end state images)?
2. The paper claims that the proposed method benefits applications in robotics and VR/AR. How is this claim substantiated? Specifically, how does the method perform in robotics-related data generation, and is there any evidence that it actually helps improve the training of embodied AI?

---

### Official Review · Reviewer_VDEv · 2025-11-01

**Soundness:** 2
**Presentation:** 2
**Contribution:** 2
**Rating:** 4
**Confidence:** 4

**Summary:**

This paper presents a way to prompt the existing image2video generation model to produce more finegrained and realistic dynamic motion. This is achieved by first creating plausible start and end frames from the text, then taking those frames to generate the final video using a proposed guided sampling. The paper also introduces a quality metric assessment system using finetuned VLM. It also introduce a synthetic dataset of around 5k videos with comprehensive quality control for finegrained motion in the videos. Overall, the proposed method seems to produce better dynamic videos than prior baselines.

**Strengths:**

1. A simple method to squeeze out more video generation with fine-grained motion capabilities from existing open-weight models
2. A controlled synthetic dataset with fine-grained motion

**Weaknesses:**

1. It is not clear that the method is scalable to produce even more dynamic video needed for training even better foundation video model. The generation ability is limited by the ability of the pretrained model.
2. It used the LLM to generate interaction and dynamic state changes. It is also limited by the generation of the LLM.
3. It is unclear whether the proposed the temporal artifact detection is helpful given that current video generation model, like Veo3, Sora2, Wan, or HunyuanVideo, are often already very realistic. It is also trained on UCF-101 which is low-res and limited in motion (vs. something newer like Kinetics or HMDB51)

**Questions:**

n/a. The authors should address the weekness above.

---

### Note · Authors · 2025-11-12

I have read and agree with the venue's withdrawal policy on behalf of myself and my co-authors.